# Auditing and Fixing Economic Validity in Tabular Foundation Models for Discrete Choice

**Yingshuo Wang** [1]  **Xian Sun** [2]  **Yanhang Li** [3]  **Zhichao Fan** [4]  **Zexin Zhuang** [5]

## Abstract

Tabular foundation models achieve strong accuracy on choice prediction tasks, but their predictions often violate the economic logic those tasks require: raising a price sometimes increases predicted demand, and implied willingness-to-pay estimates are frequently negative or implausible. We propose a two-stage adapter that embeds foundation model predictions within a utility-maximization framework. In the first stage, we estimate a standard choice model whose parameters are constrained to obey economic theory. In the second stage, we freeze those parameters and train a correction term that incorporates the foundation model's predictions as additional information. The result is a model that inherits the foundation model's accuracy gains while guaranteeing monotonic price–demand relationships under policy perturbation and producing analytically computable trade-off measures. On two transportation datasets, the adapter recovers up to 13 percentage points of accuracy over a standard logit model while maintaining perfect economic consistency, something neither the raw foundation models nor standard knowledge distillation (training a logit on TFM soft labels) achieve.

## 1. Introduction

Many consequential decisions involve choosing among discrete alternatives: which health plan to enroll in, which transit mode to take, which car to buy. Predicting these choices matters because the predictions feed directly into policy: governments set transit fares, agencies evaluate infrastructure investments, and firms set prices based on forecasted demand.

The standard approach is the multinomial logit model (MNL) and its extensions (Ben-Akiva & Lerman, 1985; Train, 2009), which derive predictions from utility maximization. This premise imposes useful structure: raising a price should lower demand, and the rate at which people trade off time against money (the value of time, or VOT) should be positive and finite. These are consequences of the assumption that choices reflect purposeful behavior.

Machine learning models consistently outperform MNL on prediction accuracy (Hillel et al., 2021), but whether they respect economic structure is less studied. Zhao et al. (2020) find that neural networks frequently violate monotonicity and produce implausible willingness-to-pay estimates. Han et al. (2022) propose architecturally constrained networks that enforce monotonicity but require redesigning the model for each application. van Cranenburgh et al. (2022) survey the field and identify economic consistency as an open challenge.

Tabular foundation models (TFMs), pretrained on large collections of tabular datasets and applied to new tasks with minimal configuration, have intensified this tension. TabPFN (Hollmann et al., 2023; 2025) performs in-context learning by treating the training set as context at inference time; Mitra (Zhang et al., 2025) scales this approach to larger datasets via AutoGluon. These models have been benchmarked on classification accuracy but not on behavioral validity. A model that correctly predicts which mode a commuter chose, but implies that higher fares *increase* ridership, would produce structurally misleading policy recommendations.

We find that this concern is not hypothetical. Auditing the TFM Mitra on two standard mode-choice datasets, Swissmetro (Swiss intercity travel) and LPMC (London urban trips), we observe monotonicity violations in up to half of test observations (only 50.8% satisfied on LPMC), negative VOT estimates in 40–65% of cases depending on mode, and nonzero probability assigned to alternatives that are physically unavailable.

The natural reaction is knowledge distillation (Hinton et al., 2015): train an MNL on the TFM's soft labels. In our

[1]University of California, Berkeley, CA, USA [2]Duke University, Durham, NC, USA [3]Northeastern University, Boston, MA, USA [4]University of Illinois Urbana-Champaign, IL, USA [5]Southern Methodist University, Dallas, TX, USA. Correspondence to: Yingshuo Wang <yingshuow@berkeley.edu>.

*Proceedings of the $2^{nd}$ ICML Workshop on Foundation Models for Structured Data*, Seoul, South Korea. 2026. Copyright 2026 by the author(s).

experiments, the distilled MNL achieves only 57.3% on Swissmetro (below the standard MNL's 63.7%) and 70.0% on LPMC (matching the MNL's 69.8%). The MNL's linear utility cannot represent the nonlinear patterns that give TFMs their accuracy advantage: the *linear* student is the bottleneck. A nonlinear but structurally constrained student could fare better; we do not pursue that route here.

We propose a different approach. Instead of replacing the TFM with a constrained model, we embed the TFM's predictions *within* a constrained model. The utility for each alternative becomes a sum of two parts: a structural component with constrained parameters that enforce economic theory, and a correction term that incorporates the TFM's predicted probabilities as additional explanatory variables. We train this model in two stages (utility parameters first, then the correction term with utility frozen) so the TFM never distorts the economic parameters.

The result is a model that (i) guarantees monotonic price–demand relationships by construction, (ii) produces analytically computable VOT from the structural parameters, (iii) assigns zero probability to unavailable alternatives, and (iv) recovers much of the TFM's accuracy advantage. On Swissmetro, the adapter gains 13 percentage points over MNL while maintaining perfect behavioral validity. On LPMC, it gains 2 percentage points, more modest but with the same structural guarantees.

## 2. Method

### 2.1. Behavioral Audit

We test three conditions that any economically valid choice model should satisfy. Each test perturbs one input variable and checks whether the model's predictions respond in the expected direction. This requires only the ability to query the model; no access to internals is needed.

**Monotonicity.** We increase the cost (or time) of one alternative by 1% of its observed range, re-run the model on the modified input, and check whether the predicted probability of choosing that alternative decreases. We report the fraction of test observations where this holds. A model grounded in utility theory scores 100%. For the adapter, the TFM predictions are pre-computed and fixed, so only the structural utility responds to the perturbation.

**Value of time (VOT).** VOT measures willingness to pay for travel time savings. In a choice model with explicit coefficients, VOT $= \beta_{\text{time}}/\beta_{\text{cost}}$; when both are negative the ratio is positive. For TFMs without coefficients, we approximate VOT via finite differences. Published benchmarks place VOT at 5–20 GBP/hr for London commuters and 50–100 CHF/hr for Swiss intercity travelers.

**Availability compliance.** A valid model assigns zero proba-

bility to unavailable alternatives. We report the mean predicted probability for unavailable options; any nonzero value is a structural error.

### 2.2. Two-Stage Behavioral Adapter

Anyone using a choice model for policy evaluation needs two things: accurate predictions (to forecast demand) and sensible economics (to evaluate interventions). A planner asking "what happens if we raise fares 10%?" needs the model to get both the level and the direction right. The MNL gives sensible economics but mediocre accuracy; the TFM gives strong accuracy but broken economics. Existing remedies do not resolve this: fine-tuning with a monotonicity penalty is impractical for in-context learners like TabPFN, and even where feasible, penalized loss does not yield interpretable coefficients. Distillation preserves economic structure but loses accuracy, as shown above. Both approaches treat the TFM and the choice model as alternatives. Instead, we put them together.

Concretely, the utility of alternative $k$ for observation $i$ is:

$$V_k(x_i) = \underbrace{V_k^{\text{struct}}(x_i)}_{\text{economic structure}} + \underbrace{\alpha \log q_k(x_i) + g_k(\mathbf{q}(x_i))}_{\text{FM correction}} \quad (1)$$

where $V_k^{\text{struct}}$ is the utility from a standard specification (alternative-specific constants, constrained time and cost coefficients, sociodemographic interactions), $q_k(x_i)$ is the TFM's predicted probability for alternative $k$, $\alpha$ is a learned scalar weight, and $g_k$ is a small neural network that maps the full TFM probability vector $\mathbf{q}$ to a residual correction for each alternative. Choice probabilities follow the standard logit formula: $P_k = \exp(V_k)/\sum_j \exp(V_j)$.

The time and cost coefficients in $V_k^{\text{struct}}$ are parameterized as $\beta = -\exp(\theta)$ for unconstrained $\theta$, guaranteeing $\beta < 0$ regardless of what the optimizer does. This makes monotonicity a mathematical certainty, not an empirical outcome: higher cost always enters the utility as a larger negative number, always reducing the predicted probability. VOT is analytically computable as $\beta_{\text{time}}/\beta_{\text{cost}}$, with no numerical differentiation needed. Availability compliance is enforced by setting $V_k = -\infty$ for unavailable alternatives.

**Why two stages?** If we train all parameters jointly, the optimizer will route information through whichever pathway minimizes loss fastest. In practice, this means the correction term $\alpha \log q_k + g_k(\mathbf{q})$ absorbs most of the signal, the structural coefficients shrink toward zero, and VOT becomes meaningless: a ratio of two near-zero numbers.

To prevent this, we train in two stages. In **Stage 1**, we freeze the correction term at zero ($\alpha = 0$, $g_k = 0$) and train only the structural parameters. This is equivalent to fitting a standard MNL. The resulting coefficients reflect economic fundamentals (time sensitivity, cost sensitivity,

mode preferences) estimated without any TFM influence. In **Stage 2**, we freeze the structural parameters and train only $\alpha$ and $g_k$. The correction term now learns to explain whatever variance the MNL left on the table, using the TFM's predictions as its information source. Because the structural parameters are frozen, the economic interpretation is locked in before the TFM touches anything.

**What each component contributes.** The TFM is good at *who chooses what*: it discriminates among alternatives at the observation level. The structural model is good at *how choices respond to policy*: its constrained coefficients encode the direction and magnitude of demand response. Because the TFM's predictions are pre-computed and fixed, the correction term acts as a per-observation intercept shift that does not alter how the model responds to cost or time changes. The structural coefficients alone determine price sensitivity, which is why monotonicity and VOT are preserved.

This mirrors standard practice: a planner perturbs features while holding everything else fixed. In the adapter, "everything else" includes the TFM's per-observation assessments; retraining on new data would re-run the full two-stage pipeline.

## 3. Experiments

### 3.1. Setup

We use two standard mode choice datasets. **Swissmetro** (Bierlaire et al., 2001) contains 10,719 stated-preference observations with three alternatives (train, Swissmetro, car) and features including travel times, costs, headway, and a binary indicator for annual travel pass (GA) ownership. We follow the standard specification: generic time and cost coefficients, with cost zeroed out for GA holders (who face no marginal cost). **LPMC** (Hillel et al., 2018) contains 81,086 revealed-preference trip records from London with four alternatives (walking, cycling, public transit, driving) and features including mode-specific travel times, costs, and sociodemographics. We use a random 10,000-observation subsample. Both datasets use 70/15/15 train/validation/test splits.

We evaluate two TFMs: **Mitra** (Zhang et al., 2025), accessed via AutoGluon, and **TabPFN v2** (Hollmann et al., 2025), which performs in-context learning at inference time. The TFMs' predicted probabilities on each split are precomputed and stored; the adapter treats them as fixed inputs. We compare against a standalone **MNL** (identical to the adapter's Stage 1) and the **raw TFM** predictions.

### 3.2. The Problem: TFMs Fail Behavioral Validity

Table 1 reports accuracy and behavioral validity for all models on both datasets. The raw TFMs achieve the highest accuracy: Mitra reaches 77.7% on Swissmetro (vs. MNL's 63.7%) and 74.2% on LPMC (vs. 69.8%). But this accuracy comes with serious behavioral failures.

On LPMC, Mitra satisfies monotonicity for only 50.8% of observations, meaning that for nearly half the test set, raising a mode's cost does *not* lower its predicted choice probability. This happens because the TFM learns correlations, not causal relationships: if higher-cost trips in the training data tend to be chosen by wealthier travelers who also have strong mode preferences, the model may associate higher cost with higher choice probability, confusing "who chooses" with "why they choose." Its finite-difference VOT estimates are negative for 40% of public transit observations and 65% of driving observations. A negative VOT implies the traveler prefers longer, more expensive trips, an economically absurd implication.

On Swissmetro, Mitra's monotonicity is higher (90.3%) but its median VOT is 0.30 CHF/hr, two orders of magnitude below published benchmarks of 50–100 CHF/hr, implying near-zero sensitivity to travel time. This would lead a planner to conclude that time savings have almost no value, a conclusion contradicted by decades of empirical evidence.

The MNL, by contrast, achieves 100% monotonicity on both datasets and produces VOT of 79.7 CHF/hr on Swissmetro (within 12% of the published benchmark of 71.1 CHF/hr from Bierlaire et al. (2001)) and 1.76 GBP/hr for London public transit and 18.3 GBP/hr for driving, consistent with the literature for short urban trips. The cost is accuracy: 14 percentage points below Mitra on Swissmetro, 4.4 points below on LPMC.

### 3.3. The Adapter Recovers Both

The two-stage adapter closes most of the accuracy gap while maintaining perfect behavioral validity (Table 1).

On Swissmetro, both Adapter+Mitra and Adapter+TabPFN reach 76.6% accuracy, recovering 13 of the 14 percentage points that separate MNL from the raw TFMs. Monotonicity is 100%, availability leak is effectively zero ($< 10^{-12}$), and VOT remains 79.7 CHF/hr. The structural coefficients are identical regardless of which TFM provides the correction (since Stage 1 trains without TFM input), confirming that the TFM does not contaminate the economic parameters.

On LPMC, Adapter+Mitra reaches 71.8% and Adapter+TabPFN 72.1%, gains of 2.0 and 2.3 percentage points over MNL, with 100% monotonicity and VOT of 1.76 GBP/hr (public transit) and 18.3 GBP/hr (driving). Across 10 different 10,000-observation subsamples,

*Table 1.* Accuracy and behavioral validity on held-out test sets. Acc: fraction of correctly predicted choices (%). Mono: fraction of observations where raising cost lowers predicted probability (%; 100 is ideal). VOT: implied value of time in local currency per hour, computed analytically for MNL and adapter models, via finite differences for TFMs. Leak: mean predicted probability assigned to unavailable alternatives on Swissmetro (%; 0 is ideal). Adapter models inherit the MNL's Stage 1 coefficients; their 100% monotonicity, near-zero leak, and VOT are therefore *guaranteed by construction*, not empirical outcomes, whereas the TFM rows are measured. † LPMC uses a 10K random subsample.

| SWISSMETRO (3 MODES, $n = 1,608$ TEST) | | | |
| --- | --- | --- | --- |
| MODEL | ACC | MONO | VOT | LEAK |
| MNL | 63.7 | 100 | 79.7 | $<.001$ |
| MITRA | 77.7 | 90.3 | 0.30 | .208 |
| TABPFN | 78.0 | — | — | — |
| ADAPT+MITRA | 76.6 | 100 | 79.7 | $<.001$ |
| ADAPT+TABPFN | 76.6 | 100 | 79.7 | $<.001$ |

| LPMC† (4 MODES, $n = 1,501$ TEST) | | | |
| --- | --- | --- | --- |
| MODEL | ACC | MONO | $\text{VOT}_{\text{PT}}$ | $\text{VOT}_{\text{DR}}$ |
| MNL | 69.8 | 100 | 1.76 | 18.3 |
| MITRA | 74.2 | 50.8 | 7.47 | $-5.7$ |
| TABPFN | 74.4 | — | — | — |
| ADAPT+MITRA | 71.8 | 100 | 1.76 | 18.3 |
| ADAPT+TABPFN | 72.1 | 100 | 1.76 | 18.3 |

Swissmetro VOT in CHF/hr (generic coefficient, scaled $\times 60$). LPMC VOT in GBP/hr: $\text{VOT}_{\text{pt}}$ = public transit, $\text{VOT}_{\text{dr}}$ = driving. Mitra LPMC $\text{VOT}_{\text{pt}}$ = 7.47 median but 40% of estimates are negative; $\text{VOT}_{\text{dr}}$ = $-5.7$ median with 65% negative. TabPFN behavioral audit omitted: its in-context learning architecture requires re-fitting on the full context for each perturbed input, making perturbation-based metrics prohibitively expensive.

the accuracy gain is $+2.2 \pm 0.5$ percentage points (95% CI: $[+1.2, +3.3]$), with all runs showing positive gains and 100% monotonicity. The learned correction weight $\alpha$ is lower on LPMC (0.41–0.44) than on Swissmetro (1.16–1.27), indicating that the MNL already captures more of the choice structure in the LPMC data. The adapter never exceeds the raw TFM's accuracy; it recovers accuracy gains, not creates them.

## 4. Discussion

**The FM correction does real work.** On Swissmetro, $\alpha$ exceeds 1.0, meaning the FM predictions are a major driver of accuracy, not a small refinement. The structural component's role is in *guarantees*, not prediction: it determines how the model responds to price and time changes, which is what enters policy analysis.

**The adapter mechanism is general.** We demonstrated

the adapter on MNL utility specifications, but the structural component can be any choice model that enforces the desired economic constraints: mixed logit for taste heterogeneity, nested logit for correlated alternatives, or neural-embedded utility models (Han et al., 2022) for flexible nonlinear specifications. The two-stage training procedure applies unchanged: fit the economic model first, then add the TFM correction. The mechanism producing behavioral constraints is the decision theory, not the dataset or the specific model form.

**Limitations.** We evaluate two TFMs on two transportation datasets. Transportation mode choice has well-established economic constraints; extending to domains with less settled theory (healthcare treatment choice, energy tariff selection) requires domain-specific audit design. The correction term's contribution varies by dataset ($+13$pp on Swissmetro, $+2$pp on LPMC), and its value depends on TFM quality; with a poor TFM, the adapter degrades gracefully to the standalone MNL. We report results from a single TFM training run per dataset; future work should examine sensitivity to TFM hyperparameters and training seeds.

## Impact Statement

This paper addresses the gap between predictive accuracy and economic validity in foundation models applied to decision-relevant tasks. Deploying models with inverted price–demand relationships could lead to misguided infrastructure investment and fare-setting decisions. Our adapter offers a practical path to combining foundation-model accuracy on factual choice prediction with the economic guarantees that policy contexts require.

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
