# OpenReview forum: "Auditing and Fixing Economic Validity in Tabular Foundation Models for Discrete Choice"
_ICML.cc/2026/Workshop/FMSD — FMSD @ ICML 2026 Poster_

### Official Review · Reviewer_UyLs · 2026-05-19
**Auditing and Fixing Economic Validity in Tabular Foundation Models for Discrete Choice**

**Rating:** 7
**Confidence:** 3

**Review:**

Summary: The paper audits two tabular foundation models (Mitra, TabPFN) on discrete choice tasks and finds pervasive economic violations: monotonicity failures in up to 50% of observations, negative willingness-to-pay, and probability leakage to unavailable alternatives. It then proposes a two-stage adapter that embeds frozen TFM predictions within a sign-constrained MNL, recovering most of the TFM's accuracy while guaranteeing monotonic price–demand relationships and analytically computable value-of-time. Evaluated on Swissmetro (+13pp over MNL) and LPMC (+2pp).


Strengths:
1. Asks the right question at the right time. TFMs are benchmarked on accuracy alone; this is the first systematic check of whether they produce actionable predictions under policy perturbation. The audit findings (e.g., 65% negative VOT for driving) are promising.

2. The adapter is elegant: embed the TFM within a constrained model rather than trying to constrain the TFM. Sidesteps the intractability of modifying in-context learners. Monotonicity is guaranteed by construction, not by a penalty that might not bind.

Very clear writing for a 4-page paper. The "who chooses what" vs. "how choices respond to policy" decomposition is effective and precise.

Areas for Improvement:

1. The frozen-TFM counterfactual assumption needs more discussion. If the TFM's accuracy comes from capturing nonlinear cost interactions, ignoring those signals during policy perturbation means the model may get the direction right but the magnitude of demand response wrong. When is this acceptable?

2. Missing baselines: (i) Han et al. (2022) is cited but not compared against; (ii) a joint-training ablation showing the claimed coefficient-collapse would strengthen the two-stage justification empirically rather than only verbally; (iii) the TabPFN behavioral audit is omitted entirely.

3. Two datasets, one domain, modest LPMC gains (+2.2 ± 0.5pp). The generalization story needs more explanation. Even a subsample-based TabPFN audit or one non-transportation dataset would help.

4. The correction network g_k is unspecified (architecture, capacity, regularization). If g_k is expressive enough to dominate the structural utility in magnitude, the economic interpretation becomes murky even if monotonicity technically holds.

5. Minor points. The distilled MNL underperforming standard MNL (57.3% vs. 63.7%) is interesting and unexplained — is this a calibration or IIA incompatibility issue? Reporting log-likelihood alongside accuracy would give a fuller picture.

Justification of Score. A well-framed workshop contribution that identifies a real problem and proposes a clean fix. The core insight transfers beyond transportation. The score is 7 because the evaluation is narrow (two datasets, one domain, one incomplete audit) and the counterfactual validity tension is acknowledged but not engaged with. Both are addressable, and the paper would spark productive discussion at FMSD.

---

### Official Review · Reviewer_MJe6 · 2026-05-20
**Clean, well-motivated fix for economic validity, but the policy-relevant counterfactual is identical to plain MNL by construction**

**Rating:** 6
**Confidence:** 4

**Review:**

The paper addresses a real and timely problem: tabular foundation models (TFMs) such as TabPFN and Mitra are accurate on discrete-choice prediction but routinely violate the economic structure those tasks presuppose. The authors first run a "behavioral audit" with three criteria, price–demand monotonicity, sign/plausibility of the implied value of time (VOT), and zero probability mass on unavailable alternatives, and document substantial failures (monotonicity satisfied for as few as 50.8% of LPMC observations under Mitra, negative VOT in 40–65% of cases, ~20% probability leak onto unavailable Swissmetro alternatives).

They then show that the obvious remedy, distilling the TFM into an MNL, underperforms because the linear student cannot represent the teacher's nonlinearities. Their proposed alternative embeds the TFM inside a constrained utility model: $V_k = V_k^{\text{struct}} + \alpha \log q_k + g_k(q)$, where $V_k^{\text{struct}}$ is a standard MNL specification with $\beta = -\exp(\theta)$ (guaranteeing negative time/cost coefficients), $q_k$ is the TFM's predicted probability, and $g_k$ is a small network on the full probability vector. Training is two-stage: fit the structural parameters first (a plain MNL), then freeze them and fit only the correction term. On Swissmetro the adapter recovers 13 of the 14 accuracy points separating MNL from the raw TFMs (76.6% vs. 63.7%), and on LPMC it adds ~2 points, in both cases with 100% monotonicity, analytic VOT, and effectively zero availability leak.

### Strengths
- Important, under-explored problem with a clean framing. "TFMs are accurate but economically incoherent" is exactly the kind of structured-data-meets-foundation-models question this workshop should host. The three-part audit is simple, query-only (no model internals), and immediately interpretable.
- The two-stage freezing idea is elegant and the right diagnosis. The observation that joint training lets the correction term cannibalize the structural coefficients (collapsing VOT to "a ratio of two near-zero numbers") is a genuine insight, and freezing Stage 1 is a clean fix that locks in interpretable economics before the TFM touches anything.
- The $\beta=-\exp(\theta)$ reparameterization turns monotonicity from an empirical hope into a structural guarantee at essentially no cost, and analytic VOT $=\beta_{\text{time}}/\beta_{\text{cost}}$ is a nice byproduct.
- Commendably honest. The paper states plainly that the adapter never exceeds the raw TFM's accuracy, that it "recovers gains, not creates them," that the contribution shrinks to +2pp on LPMC, and that it degrades gracefully to MNL with a weak TFM. The Swissmetro VOT lands within 12% of Bierlaire et al.'s published benchmark, which is a credible sanity check.

### Areas for Improvement
1. The headline accuracy gain does not transfer to the policy question that motivates the paper. Because the TFM probabilities $q(x)$ are pre-computed and frozen, when an analyst perturbs cost ($x\to x'$) the correction term $\alpha\log q_k(x) + g_k(q(x))$ is held constant; only $V^{\text{struct}}$ responds. The adapter's marginal price–demand response, the elasticity that feeds "what happens if we raise fares 10%?" is therefore identical to the plain MNL's, by construction. The TFM correction acts purely as a per-observation intercept shift that improves top-1 classification of observed choices but contributes nothing to the counterfactual response. So the paper's motivating use case (policy/fare-setting) sees no improvement over MNL, while the 13pp gain lives entirely in factual fit. The paper half-acknowledges this ("the structural component's role is in guarantees, not prediction"), but it should confront the implication head-on: in what sense does better factual accuracy help the planner, if the counterfactual demand curve is unchanged from MNL? This is the central question and currently goes unaddressed.

2. The audit, a stated contribution, only fully applies to one of the two TFMs. TabPFN's behavioral metrics are omitted because perturbation requires re-fitting the in-context model per query. That is an understandable cost argument, but it means the "audit two TFMs" claim is really "audit one TFM (Mitra) and report only accuracy for the other," which undercuts the framing.

3. Missing baselines and related work on economically constrained ML choice models. The relevant comparison is not just MNL / raw TFM / distillation. There is an established line of work on enforcing behavioral regularity in neural choice models.

4. The adapter's economic outputs add nothing over running MNL alone. VOT is "identical to MNL by construction," monotonicity is MNL's, availability is MNL's. The genuinely new object is better-fit factual probabilities with a monotone prior attached, which is valuable, but the paper sometimes frames the economic quantities as a contribution of the adapter when they are entirely Stage 1.

5. Evaluation is thin and accuracy-centric. Two transportation datasets, a single TFM training run per dataset, and top-1 accuracy as the headline metric. Choice modeling conventionally reports log-likelihood / cross-entropy, which is more informative than top-1 (especially with imbalanced mode shares) and is what distillation actually optimizes. Only LPMC has a multi-subsample CI; the marquee 13pp Swissmetro number is a single run.

### Detailed Comments / Questions
- Since $q$ is a softmax output, $\alpha \log q_k$ is (up to an additive constant) just the TFM's own logit scaled by $\alpha$; with $\alpha\approx1.2$ on Swissmetro the model is close to "TFM logits reweighted by a monotone structural prior." It would help to state this interpretation explicitly and to ablate the two correction components ($\alpha\log q_k$ vs. $g_k(q)$) to show how much each contributes.
- The "guarantee of monotonicity under policy perturbation" should be stated more carefully: it holds *because* the TFM signal is frozen at the factual input. Please make explicit that the resulting counterfactual response equals the MNL's, so readers don't over-read the guarantee as "the TFM's nonlinear price response, made monotone."
- What happens if, at perturbation time, you *do* re-query the TFM on $x'$ (the honest counterfactual)? Presumably monotonicity is no longer guaranteed — but quantifying that gap would clarify exactly what the freezing buys and costs.
- Report log-loss / negative log-likelihood alongside accuracy, and add at least one constrained/monotone-NN baseline.
- Minor: the impact statement and abstract lean on "policy validity"; given point (1), I'd soften claims that the adapter improves *policy* predictions rather than *factual* predictions with policy guarantees.

### Overall
A neat, honestly reported workshop paper on a problem that matters, with one elegant idea (stage-wise freezing + monotone reparameterization). It is held back by a conceptual gap between the factual accuracy it improves and the counterfactual policy behavior it leaves unchanged, by a narrow two-dataset/single-run evaluation, and by missing comparisons to the existing behaviorally-constrained-ML literature. These are addressable, and the framing is a good fit for FMSD, so I lean slightly positive.

---

### Official Review · Reviewer_6ecZ · 2026-05-20
**An important problem, but the manuscript needs substantial work before it is ready for an ML venue**

**Rating:** 4
**Confidence:** 3

**Review:**

## Summary:

The paper is on diagnosing and enforcing the economic validity of TFM predictions as captured by a select economic preference model. The paper finds that TFMs often violate monotonicity, underestimate the value of time, and fail to respect availability compliance. To this end, a two-stage approach is proposed that involves embedding TFM predictions within the constraint model. The experiments show that the proposed method compromises accuracy but ensures full compliance with the desired constraints.

## Strengths

The behavioral audit framework is well-motivated and operationally simple as it requires only black-box access to the model - easily applicable to any TFM without much work.

Freezing the structural parameters before introducing the TFM correction is a good idea to prevent the neural component from absorbing the economic signal.

Addresses a real deployment risk. The example of recommending raising fares to increase ridership is an important failure mode that needs to be addressed.


## Areas for improvement & detailed comments:
### Major:

Compliance is guaranteed by design, not a result. Monotonicity is enforced via $\beta = −exp(\theta)$, availability via $V_k = −\inf$ .These are architectural constraints and not empirical findings; therefore, reporting 100% monotonicity and zero availability leak in Table 1 as results is misleading because it's tautological. The interesting question is what accuracy you give up, what is the trade-off, not whether the constraints hold.

The distillation baseline and the stated student bottleneck are straw manned. Distillation into an MNL is a rather weak target, because the paper mentions neural-embedded utility models (in L212) that could preserve nonlinearity while enforcing structure. Showing that distillation into a linear model fails is not a strong argument against distillation as a general strategy.

The significance of the claims is not well supported by the empirical experiments. Results come from a single training run per dataset. Although acknowledged in the Limitations section by the authors, I strongly believe that at least a minimalistic sensitivity analysis to the random seed is required. The paper cites numerous numbers in the experiment section, yet the numbers come from a single run so their significance is questionable.

### Minor :

The paragraph that starts at L046 is logically incoherent and poorly worded. It is not specified what the stated percentages correspond to. Swissmetro and LPMC are not defined anywhere in the introduction - a reader is not aware of what they are.

Exact architecture, model size, and training details are not stated, which is a reproducibility issue.

Note that both the conference and the workshop will primarily attract an audience with an ML background, unfamiliar with economics and MNL notation. The majority of the paper is written in jargon, which is hard to understand and rather specific to economics.

Clarity or unclear jargon issues:

- L035: What does conventional distillation stand for?

- L019: "find" should be "finds"

- L024: “survey” - is used as a noun, but should be a verb

- L057–061: The distinction between "structural component" and "correction term" in Eq. (1) would benefit from a concrete example before the equation.